# Agronomic Efficiency of a New Liquid Inoculant Formulated with a Mixture of *Azospirillum brasilense* Strains Ab-V5 and Ab-V6 in Corn (*Zea mays* L.)

**DOI:** 10.3390/microorganisms13102403

**Published:** 2025-10-21

**Authors:** Ricardo Cancio Fendrich, Mayara Barbosa Silva, Ivanildo Evodio Marriel

**Affiliations:** 1Departamento de Pesquisa e Desenvolvimento, Nodusoja Indústria e Comércio Ltda, Rua Alfredo Tomacheschi, 171, Colombo 83407-330, PR, Brazil; mayara.barbosa@noduagri.com.br; 2Embrapa Milho e Sorgo, Rod MG 424 Km 45, Sete Lagoas 35701-970, MG, Brazil; ivanildo.marriel@embrapa.br

**Keywords:** inoculation, *Azospirillum brasilense* Ab-V5, *Azospirillum brasilense* Ab-V6, *Zea mays* L.

## Abstract

Nitrogen fertilization is a critical factor in maize (*Zea mays* L.) production, as nitrogen is often the primary limiting nutrient. The use of microbial biostimulants has emerged as a promising strategy to enhance nitrogen use efficiency. This study assessed the field performance of an industrially produced inoculant (Nodusoja™), formulated with *Azospirillum brasilense* strains Ab-V5 and Ab-V6, under contrasting soil and climatic conditions. The aim of this study is to assess the grain yield of maize cultivated in different edaphoclimatic conditions using the biostimulant, together with lower doses of topdressing fertilization. Field experiments were conducted across double cropping seasons in Sete Lagoas, Minas Gerais (19°28′ S; 44°15′ W), and Palmas, Tocantins (10°8′ S; 48°19′ W), Brazil, during the 2018, 2019, and 2021 harvests. Evaluated parameters included grain yield, shoot dry mass, and nitrogen content. The most pronounced effects were observed on productivity, with maximum grain yields of 8.76 and 9.05 t·ha^−1^ recorded in the 2019 season, under inoculation without topdressed N and inoculation with 50% of the recommended N dose, respectively. By contrast, uninoculated treatments with 20, 60, and 120 kg N·ha^−1^ yielded 6.41, 7.13, and 7.49 t·ha^−1^, respectively. Statistical analyses demonstrated that inoculation with strains Ab-V5 and Ab-V6 increased maize grain yield by up to 40% when combined with 50% of the recommended nitrogen fertilization. These findings highlight the potential of *Azospirillum*-based inoculants to improve N use efficiency and reduce dependence on synthetic fertilizers in maize cultivation.

## 1. Introduction

Brazilian agribusiness is one of the most important sectors of the national economy. In recent years, this sector has accounted for a significant portion of gross domestic product (GDP), contributing to the economically active labor force, and is a key component of exports, contributing to Brazil’s trade surplus. This relevance of agribusiness is due, in part, to the incorporation of modern production technologies and the occupation of new agricultural frontiers, which require the application of high doses of nutrients to increase productivity [1,2,3].

However, the intensive use of nutrients via chemical fertilizers presents low utilization efficiency by plants, increases agricultural production costs, and results in a high dependence on imported inputs, in addition to negative environmental impacts on agroecosystems [4,5].

Brazil stands out as one of the main producers and exporters of grains, with an estimated harvest of 350.2 Mt (2024/2025 harvest) [6,7]. Corn has become the second most important commodity in national agriculture, with the production of 139.7 Mt [6] in the 2024/2025 harvest (24.9 Mt in the 1st harvest (main harvest) and 112.0 of corn in the 2nd harvest (off season crop)) [7].

Agricultural production systems, particularly in the Cerrado biome, are considered to be highly dependent on fertilizers, especially nitrogen and phosphate fertilizers. The synthesis, transportation, and application of these products are energy-intensive and systematically use non-renewable and finite energy sources [8]. The synthesis of one ton of ammonia requires approximately 1.3 tons of oil, or its equivalent in fossil energy [9], and in the case of phosphate fertilizers, approximately 0.13 tons of fuel oil are consumed to produce one ton of P_2_O_5_ [10]. In addition to their costs, nitrogen fertilizers have low plant utilization efficiency, below 50% [11,12].

In this context [4], the development and adoption of biobased technologies becomes highly desirable, specifically, viable alternatives for various biotechnological processes, especially microbial inoculants [13], whether as a source of products, genes, or processes for sustainable agriculture. In the case of nitrogen, biological fixation, mediated by the enzyme nitrogenase, transforms atmospheric N_2_ into a form assimilable by plants [14], using energy from renewable sources.

The free-living bacteria of the genus *Azospirillum* interacts with plants in three ways: physically, chemically, and physiologically. The degree of each trait is strain dependent [15]. The interactions that occur with *Azospirillum brasilense* strain Ab-V5 and strain Ab-V6 are often synergistic with many mechanisms’ co-activities [16], from nitrogen fixation genes expression and biological nitrogen fixation ability [16] to root growth promotion [17] and root architecture changes induced by creating a favorable rhizosphere microbiome [18,19], auxin production, and ethylene modulation. Furthermore, it helps with the regulation of oxidative stress, indirectly inducing the oxidative stress and defense genes of maize to rise [20].

All these processes occur by colonization, adhesion, and biofilm formation, all organized through chemical communication by phytohormones synthesis, signaling, and nutrient solubilization [21]. This leads to physiological improvements via nitrogen fixation, enhancing plant stress tolerance and defense priming [22]. Thus, only by completely evaluating corn cultivation can we illustrate the contribution of microbial prospecting as a source of nutrients [23]. The provision of selected bacterial strains has allowed Brazil to save billions of dollars in foreign trading over several years [24,25,26].

Thus, the aim of this study is to assess the grain yield of maize cultivated in different edaphoclimatic conditions using, together with lower doses of topdressing fertilization, a formulated product based on a combination of the strains Ab-V5 and Ab-V6 of *Azospirillum brasilense*.

## 2. Materials and Methods

### 2.1. Preparation of Microbial Inoculants for Field Trials

The product was formulated using two strains of associative bacteria, *Azospirillum brasilense* (strains Ab-V5 and Ab-V6, (registered also as CNPSo 2083 and CNPSo 2084, respectively), licensed and obtained from Embrapa Soja, Londrina, Brazil). It was manufactured by Nodusoja Indústria e Comércio Ltda, Colombo, PR, Brazil using the following raw materials (g·L^−1^) [Êxodo Científica Química Fina Indústria e Comércio Ltda, Sumaré, Brazil]: xanthan gum, 10.0; glycerol, 10.0; yeast extract, 1.0; magnesium sulfate, 0.2, sucrose, 10.0; glucose, 2.0; ammonium chloride, 1.0; sodium chloride, 0.1; mannitol, 0.1; potassium nitrate, 0.1; carboxylic acid, 5.0; and potassium hydroxide, 5.0.

The bioprocess took place in a 3000 L fermenter [Mapan Equipamentos em Aço Inox Ltda, Farroupilha, Brazil]. First, substrates were added to an aqueous medium and pH was adjusted to 7.0. The vessel was sterilized at 121 °C for 1 h and subsequentially inoculated with a 10% rate. The broth was continuously stirred at 150 rpm at 30 °C and aerated at a rate of 0.5 vvm over 72 h.

The product was aseptically packaged in bag-in-box containers (filled with 3 L each) [Embaquim Indústria e Comércio Ltda, São Bernardo do Campo, Brazil]. The product batch was identified for concentration and shelf-life purity evaluation, as specified in the section on the quality control of inoculants based on *Azospirillum brasilense.* The product batch, approved within the guarantees, was delivered and used for all agronomic field experiments.

### 2.2. Quality Control of Inoculants Based on Azospirillum brasilense

The density of viable cells in the test inoculant was determined by the dilution method, in plates containing potato-malate medium [27], in three periods of evaluation over six months following receipt of the product, which was stored in the refrigerator. The medium composition to make the volume up to 1 L was as follows: Fresh boiled potato filtrate from 200 g of *Solanum tuberosum* (approximately 0.8 L); raw material (g·L^−1^): malic acid, 2.5; sucrose, 2.5; potassium hydroxide for pH adjustment (between 6.5 and 7.0) [Carbon Científica Ltda, São José dos Pinhais, Brazil]; bacteriological agar, 10.0 [Kasvi, São José dos Pinhais, Brazil]; nutrients (mg·L^−1^) CuSO_4_·5H_2_O, 0.08; ZnSO_4_·7H_2_O, 2.4; H_3_BO_3_, 2.8; Na_2_MoO_4_·2H_2_O, 2.0; MnSO·H_2_O, 2.35; biotin, 0.1; pyridoxine-HCL, 0.2 [Biotecc Comércio de Produtos para Laboratório, Curitiba, Brazil].

From aliquots of each sample, aliquots of decimal serial dilutions (10^−1^ to 10^−7^), in duplicates, were transferred to the culture medium and left for five days of incubation at 28 °C. After this period, the colonies were counted, and the cell concentration was expressed in Colony Forming Units (CFU·mL^−1^). The growth rate expected for *Azospirillum* in this medium is between 3 and 5 days, according to its metabolism and colony morphology, as described by Döbereiner et al. [27]. Furthermore, this period is a normative recommendation, in order to prove that there is no contaminant.

### 2.3. Characteristics of Experimental Areas

The field experiments were conducted at Embrapa Milho e Sorgo, located in Sete Lagoas, Minas Gerais State (MG), Brazil (latitude: 19°28′ S and longitude: 44°15′ W; 761 m altitude), at Embrapa Arroz e Feijão, in Santo Antônio de Goiás (GO), Goiás State, Brazil (latitude: 16°28′ S and longitude: 49°17′ W; 766 m altitude), and at Embrapa Pesca e Aquicultura, in Palmas, TO, Brazil (latitude: 10°8′ S, longitude: 48°19′ W). The climate in Sete Lagoas is a tropical highland, monsoon-influenced, humid, subtropical climate classified by Köppen as Cwa [28], with an annual rainfall of 1382.7 mm [29], an altitude of 794 m, and an average annual temperature of 21.6 °C. The climate of Santo Antônio de Goiás is a tropical savannah, megathermal climate, classified by Köppen as Aw, with an altitude of 799 m and an average annual temperature of 23.1 °C. In Palmas, the climate is also Aw, tropical, with an average annual temperature of 26.7 °C, an altitude of 409.9 m, and an average annual rainfall of 1760 mm. The municipality has a well-defined rainfall pattern, characterized by a rainy season from October to April and a dry season from May to September, with an average annual rainfall of 1472.8 mm and 70% relative humidity.

In Sete Lagoas (MG), the experiments were conducted in two environments: Cerrado (dystrophic Dark Red Latosol, with a clayey texture) and Várzea (hydromorphic with a clayey texture). In Palmas (TO), experiments were carried out in an agricultural soil with constructed fertility, also classified as Dark Red Latosol, with a clayey texture, in sub perennial Cerrado.

The chemical characteristics of the soils are represented in Table 1. All field tests carried out to assess agronomic viability and efficiency were executed in accordance with the provisions of the annex to Normative Instruction SDA 13, of 25 March 2011 [30]. An exception was the experimental areas of Santo Antônio de Goiás, Goiás, which were dismissed from the results and discussion due to unfavorable climatic events.

### 2.4. Experimental Design, Treatments and Trial Conduct

The experiments were conducted over three growing seasons, in the 2017/2018, 2018/2019, and 2020/2021 agricultural years, the latter being the off-season. The experiments were conducted in a randomized complete block design with four replicates. Each plot consisted of six 6 m-long rows, spaced 0.7 m between rows and 0.20 m between plants, with the useful plot having four central rows 4 m long.

A basic planting fertilizer consisting of 500 kg·ha^−1^ of the 4-30-16 formula and 20 kg·ha^−1^ of slow-release micronutrient (FTE BR12) were used. The total applied amounts were 20 kg·ha^−1^ N, 150 kg·ha^−1^ P_2_O_5_, and 80 kg·ha^−1^ K_2_O, in the forms of urea, triple superphosphate, and potassium chloride, respectively. Nitrogen fertilization was applied as follows: 1/3 at planting and 2/3 as topdressing [Nutriplant Indústria e Comércio de Fertilizantes Ltda, Barueri, Brazil].

The following treatments, resumed in Table 2, were tested: (i) control, without nitrogen topdressing, without inoculation (N20-unin); (ii) 20 kg·ha^−1^ at planting + 40 kg·ha^−1^ of N at topdressing, equivalent to a half dose of N, without inoculation (N60-unin); (iii) 40 kg·ha^−1^ at planting + 80 kg·ha^−1^ of N at topdressing, equivalent to one dose of N, without inoculation (N120-unin); (iv) Nodusoja™ inoculant, in the absence of N at topdressing (N20- Nodu); (v) Nodusoja™ inoculant, in the presence of 20 kg·ha^−1^ at planting + 40 kg·ha^−1^ N, a half dose of N (N60-Nodu); (vi) standard commercial inoculant, in the absence of N at topdressing (N20-Stand); and (vii) standard commercial inoculant, in the presence of 20 kg·ha^−1^ at planting + 40 kg·ha^−1^ N at topdressing (N60-Stand).

### 2.5. Application of Inoculants to Seeds

The liquid inoculant tested was Nodusoja™, formulated with a mixture of *Azospirillum brasilense* strains Ab-V5 and Ab-V6. The standard liquid biostimulant containing strains Ab-V5 and Ab-V6 was purchased locally.

Inoculation was carried out by applying the products to the seeds with the use of a calibrated micropipette, just before planting, far from insecticides or fungicides mixtures and according to the manufacturers’ recommendations (at a dose of 100 mL·ha^−1^); i.e., normalizing 20 billion CFU per hectare, adjusting the volume as necessary.

### 2.6. Evaluated Parameters

Corn grain yield (t·ha^−1^) was evaluated in extended trials during the 2018 and 2019 harvests in Sete Lagoas and 2021 in Palmas. The N content of recently mature leaves (entire leaf blade, excluding the midrib) was sampled at the R5 dent stage, in order to reflect nitrogen status related to grain yield. It was evaluated by semi-micro Kjeldahl methodology, as described by Carmo et al. [31]. In this case, four plants were collected per plot, washed, weighed, ground, and dried to estimate dry matter content and then digested for N content and percentage of N.

The instrumentation for nitrogen analysis was performed using a semi-analytical balance (BG 400) [Gehaka Ltda, São Paulo, Brazil], a digestion block (15-50) [Sarge Corp, Potomac, MD, USA] with temperature controller (TE-007MP) [Tecnal, Piracicaba, Brazil], an automatic titrator (ABU 80, TTT 80, PHM 82, TTA 80) [Radiometer Medical ApS, Copenhagen, Denmark], a fume hood [Grupo Vidy, Taboão da Serra, Brazil], and a Kjeldahl nitrogen distiller (TE-036) [Tecnal, Piracicaba, Brazil].

### 2.7. Statistical Analysis

The data were subjected to normality and homogeneity using variance tests. Next, analysis of variance (ANOVA) and the Scott–Knott test were performed, in a general way, to compare means, with a 10% significance level. All calculations were performed using SISVAR software, version 5.6 (disponible at https://des.ufla.br/~danielff/sisvar.html#download, accessed on 14 October 2025) [32].

## 3. Results

### 3.1. Quality of Products Formulated Based on Azospirillum brasilense

The results of Nodusoja™-formulated products based on Ab-V5 and Ab-V6 used in different field experiments resulted in the following average values of three subsamples from each period evaluated: Sete Lagoas (second harvest, 2018) 3.0 × 10^8^ CFU·mL^−1^, Sete Lagoas (second harvest, 2019) 5.0 × 10^8^ CFU·mL^−1^, and Palmas, TO (second harvest, 2021) 2.0 × 10^8^ CFU·mL^−1^.

### 3.2. Corn Grain Productivity

The results obtained for grain productivity among the trials conducted in Sete Lagoas are presented in Table 3. Significant differences (*p* < 0.05) were observed between the treatments tested in the four trials, regardless of the year and environment, with values ranging from 4.87 to 9.05 t·ha^−1^.

In the 2021 harvest, evaluated in Tocantins, there were also significant differences between treatments (*p* < 0.10), with grain yields ranging from 6.00 to 8.03 t·ha^−1^ (Table 4). In this case, no response to inoculation was observed, except for the treatment with a standard microbial biostimulant in the presence of 40 kg·ha^−1^ of N, which was 17% higher than the uninoculated control, but equivalent to the uninoculated nitrogen fertilization, with 40 kg·ha^−1^ of N as a topdressing (N60-unin).

On the other hand, a significant response to the applied N levels was observed in Palmas, Tocantins, Brazil (Table 4).

### 3.3. Nitrogen Content in Corn Plants

The results obtained in the trials conducted in the 2018 and 2019 harvests, in the Cerrado and Várzea—Sete Lagoas, for plant N content, are presented in Table 5. In the 2018 harvest, significant differences (*p* < 0.10) were observed only in the case of grain N content, with different grouping between inoculated and uninoculated treatments. In this case, regardless of the products, plants inoculated in the presence of N showed greater N accumulation when compared to the uninoculated treatments; however, means did not differ from each other within the group. And in the case of the Cerrado 2019 harvest, two treatments that did not receive topdressing N had slightly smaller leaf N content than the others.

## 4. Discussion

### 4.1. Formulated Product Based on Azospirillum brasilense Ab-V5 and Ab-V6

The type of formulation developed played a fundamental role in the field results, as morphological and physiological differences in the biostimulant can affect its operational viability. Details and challenges are also discussed by Figiel et al. [33] and Nakatani et al. [34].

Quality monitoring of the Nodusoja™ test product based on Ab-V5 and Ab-V6 was evaluated based on the serial dilution method over six months and revealed a viable cell density quantified at 2.0 × 10^8^ (CFU·mL^−1^), in accordance with Normative Instruction 30 of 12 November 2010 [35] for industrial batches of biostimulant production.

The search for sustainable technologies that efficiently meet agricultural demand and maintain production standards is growing. Therefore, the use of nitrogen-fixing bacteria in corn, such as *Azospirillum brasilense* (strains Ab-V5 and Ab-V6), is already widely known in Brazilian agriculture and is recognized as a reliable biostimulant for corn crops, as thoroughly described by Hungria & Nogueira [2]. Among the many positive effects of their application, the stimulation of root development by phytohormones stands out, phytohormones being largely responsible for increased corn productivity and, consequently, for the reduction in the use of nitrogen fertilizers [24]. However, in order to find new, increasingly efficient formulations capable of increasing productivity and reducing production costs, biostimulants remain a source of study [36,37,38].

Currently, the inoculant industry in Brazil already represents a growing and significant portion of the competitive global agro-industrial market, as a substitute or complement to nitrogen fertilization, in the case of grasses [39], when applied alone [40], or in soybean crops in co-inoculation with rhizobia [41].

### 4.2. Discussion on Yield Measurements

In the 2018 harvest, in a Cerrado environment, an increase in productivity was observed due to inoculation, when compared to the treatment without nitrogen fertilization and without inoculation (negative control), for both tested products, with higher average values of around 45% (Table 3). However, grain yield in the presence of the test biostimulant (Nodusoja™) based on Ab-V5 and Ab-V6 was similar to or higher than the standard treatment (positive control), a commercial microbial biostimulant (Masterfix Gramíneas, manufactured by Stoller do Brasil), regardless of N availability.

However, significant productivity gains were observed with combined inoculation with half the recommended dose of N (N60), with increases of around 23% and 17% for the (N60-Nodu) and (N60-Stand) products based on Ab-V5 and Ab-V6, equivalent to +1.68 and +1.2 tons·ha^−1^, respectively, in relation to the uninoculated treatment (N60-unin). It is worth noting that, in both cases, inoculation provided productivity gains higher than those observed with the isolated application of 120 kg·ha^−1^ of N, which did not differ statistically from the dose of 60 kg·ha^−1^ of N.

In the trials conducted in the floodplain, 2018 harvest, the behavior of the treatments was generally similar to that observed in the Cerrado (Table 3). The treatment with inoculation of the test product Nodusoja™ based on Ab-V5 and Ab-V6 provided significant gains in grain yield compared to the negative control treatment (N20-unin), with values ranging from 27 to 37% (extra +1.8 to +2.4 tons·ha^−1^), depending on the product, which did not differ from each other when in the presence of 40 kg·ha^−1^ of N as topdressing (N60). These results were similar to those observed with the isolated dose of 120 kg·ha^−1^ of N, although there was a productivity response as a function of N availability.

In the 2019 harvest, regardless of the environment, the productivity of corn inoculated with the Nodusoja™ product was statistically superior to the uninoculated treatment, without inoculation and without N topdressing (N20-unin), with increases of 36% (+2.3 tons·ha^−1^), and also presented superior results to treatments with the microbial biostimulant used as positive control at the two N levels tested, in the soils of Cerrado biome and Várzea (floodplain).

Positive effects on grain yield promoted by inoculation with *A. brasilense* have already been described by different authors [39,42]. In a study carried out with corn and different strains of bacteria, Hungria et al. [39] found increases of up to 30% when compared to the control treatment. Lana et al. [42] found increases of 7% to 14% and 29% in studies in the Cerrado region. Considering the experiments carried out with the Nodusoja™ biostimulant, the increases in corn yield were equivalent to those observed by these authors. Also, there were positive effects comparable to the use of other types of formulation, such as those described in Garcia et al. [43].

In the soils of Sete Lagoas, corn yield results were more promising than those observed in Tocantins. Soil and climate characteristics may be the main factors causing discrepancies in the results obtained in the experiments, but there are indications that the Nodusoja™ microbial product based on Ab-V5 and Ab-V6 has the potential to be equivalent to agronomic controls.

According to data obtained by Szilagyi-Zecchin et al. [40], who evaluated the efficiency of applying microbial inoculants formulated with *A. brasilense* in corn crops cultivated with different nitrogen doses under field conditions, the average productivity results of the four locations were 34% when comparing the best treatment.

### 4.3. Discussion on Differences in Nitrogen Levels

In most cases, the nitrogen content had no response when nitrogen addition was taken into consideration. On the other hand, in the 2019 harvest, significant differences were observed only for the data from the trial conducted in the Cerrado (Table 5). In general, plants inoculated in the presence of N and N alone showed greater nutrient incorporation, compared to the uninoculated control, without N topdressing. Except for the treatment with the Nodusoja™ inoculant, without N topdressing, in which the foliar N content was similar to the uninoculated control at the 2019 harvest at Cerrado. These overall data suggest that greater effects of inoculation occur on the efficiency of Nitrogen translocation from shoots to grains.

However, it is worth noting that the efficiency of plant × bacteria interactions depends on the genotype of the macro and microsymbiont, in addition to the product used and abiotic effects [44].

In the case of non-legumes, following the pioneering discoveries of the association of these plants with diazotrophic bacteria [45,46], the genus *Azospirillum* spp. has become widely studied, with more than a dozen species already described, obtained from diverse environments around the globe. The beneficial effects of this bacterium on plant growth involve multiple mechanisms, such as biological fixation of atmospheric nitrogen, production of phytohormones, enzymes of the Krebs and nitrogen cycles, among others [47,48]. Several literature review works have listed and discussed in detail the scientific and practical advances, in recent decades, on inoculation technology with microorganisms, as well as its economic viability, with productivity gains of up to 30% [49,50].

### 4.4. Closing Remarks

This study observed that in Sete Lagoas, on Cerrado soil, inoculation with the formulated product Nodusoja™ was equivalent or superior to the application of 60 kg·ha^−1^ of nitrogen fertilizer. This observation can also be extrapolated from data from the floodplain, although it is safer to state that the application of the Nodusoja™ inoculant was equivalent to the application of 60 kg N·ha^−1^. Therefore, industrial fertilizer should not be completely replaced by the inoculant in grasses, since biological nitrogen fixation by microorganisms contributes only small portions [51]. The arrangement of all *Azospirillum* spp. functions to promote plant growth may result in plants with a greater capacity to absorb nutrients and water from the soil, improving nutrition and growth [41,47,52]. Another important factor is that the use of biostimulants based on *A. brasilense* Ab-V5 and Ab-V6 can generate significant savings in industrial fertilizers, making it possible to reduce transportation costs and pollution related to the production and application of mineral fertilizers.

The use of rigorously selected microorganisms as a strategy for greater efficiency in nitrogen utilization by corn plants, as detailed above, has numerous advantages beyond biological ones. In this case, reducing the use of nitrogen fertilizers is a potential advantage in the medium and long term, which can lead to lower production costs and a reduced environmental impact. It is known that biostimulants based on *Azospirillum* sp. can cause physiological and morphological changes in the root system during plant growth, which can indirectly or directly lead to changes in the structure and diversity of root-associated bacterial associations [53].

On the other hand, maintaining and/or increasing the competitiveness of national agricultural businesses, both internally and externally, depends on technological innovations that allow for the sustainability of these activities, with the insertion of new parameters such as reducing production costs, reducing the consumption of fossil fuels, and reducing the negative environmental impacts of the technologies used [4].

Several factors contribute to maintaining the quality of an inoculant, ensuring consumer confidence in its effectiveness in the field [33]. Because changes in the morphological and physiological characteristics of cells during inoculant production can affect their survival, it is important to develop a commercial formulation that is effective in maintaining cell viability for storage periods compatible with the production and distribution process, as well as its operational viability [54,55].

In general, the results demonstrate that the practice of inoculating corn with atmospheric nitrogen-fixing bacteria of the genus *Azospirillum brasilense* (strains Ab-V5 and Ab-V6), in an appropriate formulation, provides an increase in grain productivity, regardless of nitrogen availability in the soil.

## 5. Conclusions

The combination of the strains showed the stability and capacity of root colonization, demonstrating the biological contribution of the biostimulants to maize crops under different field conditions. The use of appropriately formulated microbial biostimulants combined with 60 kg·ha^−1^ of N can provide corn grain yield gains of up to 40% compared to cultivation with 20 kg·ha^−1^ of N but without inoculation, depending on the chemical characteristics of the soil and the product used. Grain yield increases were observed in 80% of the cases evaluated.

The contribution of inoculation with the microbial inoculant of *Azospirillum brasilense* Ab-V5- and Ab-V6-based formulation Nodusoja™ generates an increase in productivity, probably through greater efficiency in the use of nitrogen by plants, via its translocation from the aerial part to grains.

The Nodusoja™ microbial inoculant test, based on Ab-V5 and Ab-V6, presents agronomic efficiency superior to or equivalent to the standard commercial product tested, with an increase of up to 60%, depending on the agricultural year and soil nitrogen availability. Greater efficiency was observed in the presence of half the N dose recommended for the region, 60 kg·ha^−1^ of N.

The results presented reinforce the widespread and well-recommended use of the microbial inoculant based on an Ab-V5 and Ab-V6 mixture as a strategy to increase corn productivity, particularly in corn cultivation in the presence of low and medium nitrogen availability in the soil.

## Figures and Tables

**Table 1 microorganisms-13-02403-t001:** Soil profile and chemical analysis at the sites; samples taken prior to the trials where the field experiments were conducted.

Location	Sete Lagoas, MG 2018/2019		Palmas, TO 2020/2021
	Cerrado Soil	Várzea (Floodplain)	
pH_H2O_	6.2	6.2	5.8
P_Mehlich-1_ (mg dm^−3^)	46.9	58.2	4.0
K (mg dm^−3^)	106.8	122.7	4.6
Ca (cmol_c_ dm^−3^)	5.40	6.33	1.3
Mg (cmol_c_ dm^−3^)	1.18	0.76	0.7
Al (cmol_c_ dm^−3^)	0.00	0.00	0.00
H + Al (cmol_c_ dm^−3^)	1.70	1.80	1.9
SB * (cmol_c_ dm^−3^)	6.86	7.40	2.1
V ** (%)	80.10	80.40	52
SOM *** (dag kg^−1^)	3.91	1.38	1.7

* SB: Sum of exchangeable bases (SB); ** V: Base saturation; *** SOM: Soil organic matter.

**Table 2 microorganisms-13-02403-t002:** Description of treatments used in the experiments.

Treatments	Description
N20-unin	20 kg·ha^−1^ N in planting + without topdressing N fertilizer application
N60-unin	20 kg·ha^−1^ N in planting + 40 kg·ha^−1^ N topdressing fertilization
N120-unin	40 kg·ha^−1^ N in planting + 80 kg·ha^−1^ N topdressing fertilization
N20-Nodu	Formulated microbial inoculant Nodusoja™ based on *Azospirillum brasilense* strains Ab-V5 and Ab-V6, without topdressing N fertilizer application
N60-Nodu	Formulated microbial inoculant Nodusoja™ based on *Azospirillum brasilense* strains Ab-V5 and Ab-V6 + 40 kg·ha^−1^ N topdressing fertilization
N20-Stand	Standard commercial microbial inoculant based on *Azospirillum brasilense* strains Ab-V5 and Ab-V6, without topdressing N fertilizer application
N60-Stand	Standard commercial microbial inoculant based on *Azospirillum brasilense* strains Ab-V5 and Ab-V6 + 20 kg·ha^−1^ N in planting + 40 kg·ha^−1^ N topdressing fertilization

**Table 3 microorganisms-13-02403-t003:** Average productivity (t·ha^−1^) of corn grains in Sete Lagoas, in Cerrado biome and Várzea (floodplain) soil, in the 2018 and 2019 harvest years.

Treatments *	2018 Harvest		2019 Harvest	
	Cerrado	Várzea	Cerrado	Várzea
N20-unin	5.014 c	5.593 b	6.414 b	5.433 b
N60-unin	6.119 b	5.885 b	7.133 b	5.237 b
N120-unin	6.533 b	7.113 a	7.495 b	7.130 a
N20-Nodu	7.004 a	6.247 b	8.765 a	7.067 a
N60-Nodu	7.518 a	7.136 a	9.048 a	7.977 a
N20-Stand	6.416 b	7.669 a	6.665 b	5.696 b
N60-Stand	7.164 a	7.586 a	6.889 b	4870 b
CV **	8.65	8.58	14.97	13.87

* Averages followed by the same letter do not differ statistically from each other at 5% according to Scott-Knott’s test, grouped by harvest season and location. ** (CV—statistical coefficient of variation).

**Table 4 microorganisms-13-02403-t004:** Corn grain productivity (t·ha^−1^) in Palmas, TO, 2021 harvest.

Treatments *	2021 Harvest
	Palmas, TO
N20-unin	6.001 c
N60-unin	6.978 b
N120-unin	8.030 a
N20-Nodu	6.494 c
N60-Nodu	6.477 c
N20-Stand	5.979 c
N60-Stand	7.041 b
CV **	9.87

* Averages followed by the same letter do not differ statistically from each other at 10% according to Scott-Knott’s test. ** (CV—statistical coefficient of variation).

**Table 5 microorganisms-13-02403-t005:** Leaf and grain nitrogen (N) content in corn grown in Sete Lagoas-MG, in Cerrado biome and Várzea (floodplain) soil, in the 2018 and 2019 harvest years.

	2018 Harvest			2019 Harvest	
Treatments	Cerrado Soil		Várzea (Floodplain)	Cerrado Soil	Várzea (Floodplain)
	Leaf N	Grain N	Leaf N	Leaf N	Leaf N
	(kg·ha^−1^)	(kg·ha^−1^)	(kg·ha^−1^)	(kg·ha^−1^)	(kg·ha^−1^)
N20-unin	146.56 a	75.77 b	163.28 a	128.61 b	235.62 a
N60-unin	135.18 a	90.03 b	220.11 a	196.00 a	193.24 a
N120-unin	172.35 a	87.40 b	233.64 a	235.19 a	235.29 a
N20 + Nodu	183.66 a	102.36 a	210.62 a	136.35 b	216.23 a
N60 + Nodu	149.99 a	108.99 a	279.60 a	184.28 a	281.07 a
N20 + Stand	158.00 a	96.01 a	292.06 a	173.01 a	236.49 a
N60 + Stand	187.91 a	108.46 a	271.44 a	193.91 a	247.65 a

Compared means within a single column followed by the same letter do not differ statistically from each other at 10% according to Scott-Knott’s test, grouped by harvest season, location, and variable.

## Data Availability

Data could only be disclosed after technical opinion was published as an Embrapa field experiments report, including the agronomic field results of preliminary experiments needed for the evaluation. The favorable technical opinion (number 244/2023/SISV-PR/DDA-PR/SFA-PR/SE/MAPA) was released by the Ministério de Pecuária e Abastecimento MAPA, which allowed for the validation of the product registration to be released after experimentation. The data obtained can be published, as they have already been approved after evaluation, and a favorable opinion to carry out tests with these products has been released. No part of the data has ever been publicly submitted before, and thus, it is a novel scientific contribution.

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
