# Peer review of "Agronomic Efficiency of a New Liquid Inoculant Formulated with a Mixture of *Azospirillum brasilense* Strains Ab-V5 and Ab-V6 in Corn (*Zea mays* L.)"

_microorganisms, 2025, doi:10.3390/microorganisms13102403_

Round 1

Reviewer 1 Report

Comments and Suggestions for Authors

Specific comments for authors:

Introduction

  1. Include additional background on the agricultural application of Azospirillum strains. Mention previous research where these bacteria have been used as plant growth-promoting agents, as an alternative to conventional chemical fertilizers. Include one or two sentences describing their mechanisms of action in plants.
  2. Clearly state the aim and hypothesis of the study as the final paragraph of the Introduction section. This will clarify the study’s purpose and allow readers to assess its scientific significance.

Materials and Methods

  1. Preparation of microbial inoculants for field trials: Clarify whether Nodusoja™ was developed for the first time in this study or if it is a commercially available product. If it is commercial, explain the rationale for describing its production process. If the study only evaluates the efficacy of an existing commercial product, the scientific rationale is unclear. Using a commercial product as a positive control is understandable, but testing its efficacy alone lacks novelty.
  2. For these reasons, explicitly state the research objective at the end of the Introduction.
  3. Specify what Nodusoja™ is intended for according to the manufacturer’s specifications. If the product is intended for a different plant species than maize, testing it on maize could provide a meaningful insight. Otherwise, the study lacks novelty.
  4. Lines 81–82: As Dr. Ivanildo Evodio is an author, there is no need to specify that he conducted the experiment; this should be included in the Author Contributions section.
  5. Line 170 – Evaluated parameters: Specify the timing of harvests and yield determination. Describe the method used for determining nitrogen content, including the instrument and plant part analyzed.

Results

  1. Line 186 – Quality of products formulated based on Azospirillum brasilense: This section seems to describe methods rather than results and should be moved to Materials and Methods.
  2. Tables 3 and 4: Add units (kg ha⁻¹) to all relevant columns.

Discussion

  1. The Discussion largely repeats results obtained in this study. It should be expanded to include comparisons with other studies that used Azospirillum Provide more literature-based context to support the findings.

Additional Comments

  1. For studies of this type, include images of the field trials. Photos showing the trial setup and potential differences between treatments would be valuable.
  2. While tables provide clarity, for a journal of this level, consider visual representation of results (e.g., bar charts) to make differences among treatments more immediately apparent.

Author Response

Comments 1: [Include additional background on the agricultural application of Azospirillum strains. Mention previous research where these bacteria have been used as plant growth-promoting agents, as an alternative to conventional chemical fertilizers. Include one or two sentences describing their mechanisms of action in plants.]

Response 1: Thank you for pointing this out. Therefore, we have improved the excerpt accordingly. The text was updated in the manuscript.

Comments 2: [Clearly state the aim and hypothesis of the study as the final paragraph of the Introduction section. This will clarify the study’s purpose and allow readers to assess its scientific significance.]

Response 2: We appreciate you bringing this to our attention. We fully concur with this observation. The text was updated in the manuscript.

Comments 3: [Preparation of microbial inoculants for field trials: Clarify whether Nodusoja™ was developed for the first time in this study or if it is a commercially available product. If it is commercial, explain the rationale for describing its production process. If the study only evaluates the efficacy of an existing commercial product, the scientific rationale is unclear. Using a commercial product as a positive control is understandable, but testing its efficacy alone lacks novelty.]

Response 3: Your feedback is highly valued. Permit us to provide further insight. For better understandind we changed the term to differentiate better. The earlier described as "commercial avaiable product" from now on will be described as "trial-approved-product", since Nodusoja™ was developed for the first time in this study. This explains many misunderstanding points. The use Azospirillum, altought is common and safe, agronomic proofing of the mode of application is essential to scientificaly validate the use of the combination of the strains, so it can be reproducible without drawbacks, allowing the agricultors benefits from its the use.

Comments 4: [For these reasons, explicitly state the research objective at the end of the Introduction.]

Response 4:  The text was updated in the manuscript.

Comments 5: [Specify what Nodusoja™ is intended for according to the manufacturer’s specifications. If the product is intended for a different plant species than maize, testing it on maize could provide a meaningful insight. Otherwise, the study lacks novelty.]

Response 5: This is a valuable observation; however, our analysis leads us elsewhere. We’d be happy to elaborate an explanation.

Comments 6: [Lines 81–82: As Dr. Ivanildo Evodio is an author, there is no need to specify that he conducted the experiment; this should be included in the Author Contributions section.]

Response 6: Thank you. The item has been placed in its designated section of the manuscript.

Comments 7: [Line 170 – Evaluated parameters: Specify the timing of harvests and yield determination. Describe the method used for determining nitrogen content, including the instrument and plant part analyzed.]

Response 7: Thank you. Accordingly, we have detailed further this item in the text.

Comments 8: [Line 186 – Quality of products formulated based on Azospirillum brasilense: This section seems to describe methods rather than results and should be moved to Materials and Methods.]

Response 8: Your remarks have been duly noted and appreciated. We have relocated the passage to its appropriate position within the manuscript.

Comments 9: [Tables 3 and 4: Add units (kg ha⁻¹) to all relevant columns.]

Response 9: Thank you for your attention to detail. It has been considered. Also, units have been changed as other reviewer solicitation.

Comments 10: [The Discussion largely repeats results obtained in this study. It should be expanded to include comparisons with other studies that used Azospirillum Provide more literature-based context to support the findings.]

Response 10: Your comments are acknowledged with sincere appreciation. Accordingly, we have discussed further this item in the text.

Comments 11: [For studies of this type, include images of the field trials. Photos showing the trial setup and potential differences between treatments would be valuable.]

Response 11: We value your input on this matter. But unfortunately we could not fulfill with this request. Many experiments have been performed at the covid-19 period and fewer people worked, no one for obtaining photos.

Comments 12: [While tables provide clarity, for a journal of this level, consider visual representation of results (e.g., bar charts) to make differences among treatments more immediately apparent.]

Response 12: Thank you for your valuable feedback. We generally support the perspective shared. We’d be happy to elaborate an explanation. This was updated in the manuscript.

Reviewer 2 Report

Comments and Suggestions for Authors

Dear authors,

A big work has been done on nitrogen metabolism in grains and field experiments have been conducted. The topic of current manuscript is highly interesting. Nevertheless, in my opinion, if authors want to publish this work in Microorganisms journal, more attention should be paid on studied bacteria - Azospirillum brasilens. What are the mechanisms of interaction? What are the in vitro PGP-properties of these strains? 

INTRODUCTION

I didn’t like the introduction section much, since it underlines the economic value of agriculture in Brazil. Please get to know aims and scope of Microorganisms journal and rewrite your introduction facing it towards the microbes. 

Line 41 - its better to change the word «presents» to «results in»

Paragraph (lines 44-52) about grain production in Brazil should be shorten.

Line 57-58 - Please use the correct reference format

Line 63 - delete the word «for»

Lines 66-68 - change the word «become» to «becomes» and delete the word «demonstrably»

It would be nice to write the aim of your study in the introduction section. 

MATERIALS AND METHODS

Lines 80-83 - The sentence «Agronomic efficiency tests of liquid inoculants were…» should be moved to ACKNOWLEDGEMENTS.

What is the source of Azospirillum brasilense (strains Ab-V5 and Ab-V6)? How these strains were isolated and identified? What is the difference between them? Why using both? 

Lines 87-94 - Please rewrite the sentence: «The bioprocess took place in a 3000L fermenter, as the following steps: (i) substrates were added to an aqueous…» making making separate, coherent, and sequential sentences with a subject and a predicate. Since it is very difficult to understand overwise.  

Line 99 - specify the potato-malate medium 

Why CFU were counted after 5 days post plating? Usually we calculate the CFU after 24 hours. Provide the ref or explain please.

Lines 131-133 - «All field tests car- 131 ried out to assess agronomic viability and efficiency were carried out in accordance with 132 the provisions of the annex to Normative Instruction SDA 13, of March 25, 2011» - correct the sentence, «carried out» is repeated twice. 

How was N-content measured?

RESULTS

Table 3, Table 4 - decipher what СV means.

Results section is very compressed. Please describe obtained results more detailed

Table 5 - why grain nitrogen (N) content is showed only for Cerrado soil 2018 variant?

Author Response

Comments 13: [Line 41 - its better to change the word «presents» to «results in»]

Response 13: Thank you for your attention to detail. The text was updated in the manuscript.

Comments 14: [Paragraph (lines 44-52) about grain production in Brazil should be shorten.]

Response 14: Your remarks have been duly noted and appreciated. Thus, it has been revised. The text was updated in the manuscript.

Comments 15: [Line 57-58 - Please use the correct reference format]

Response 15: Thank you for your attention to detail. This was updated in the manuscript.

Comments 16: [Line 63 - delete the word «for»]

Response 16: Thank you for your attention to detail. The text was updated in the manuscript.

Comments 17: [Lines 66-68 - change the word «become» to «becomes» and delete the word «demonstrably»]

Response 17: Thank you for your attention to detail. The text was updated in the manuscript.

Comments 18: [It would be nice to write the aim of your study in the introduction section.]

Response 18: We thank you for your professional perspective. We acknowledge the validity of this point. Accordingly, we have detailed further this item in the text.

Comments 19: [Lines 80-83 - The sentence «Agronomic efficiency tests of liquid inoculants were…» should be moved to ACKNOWLEDGEMENTS.]

Response 19: Thank you for highlighting this issue. We have relocated the passage to its appropriate position within the manuscript.

Comments 20: [What is the source of Azospirillum brasilense (strains Ab-V5 and Ab-V6)? How these strains were isolated and identified? What is the difference between them? Why using both?]

Response 20: Your comments are acknowledged with sincere appreciation. This is a very important point. Accordingly, we have discussed further this item in the text. (discussion section)

Comments 21: [Lines 87-94 - Please rewrite the sentence: «The bioprocess took place in a 3000L fermenter, as the following steps: (i) substrates were added to an aqueous…» making making separate, coherent, and sequential sentences with a subject and a predicate. Since it is very difficult to understand overwise.]

Response 21: We acknowledge the validity of this point. Therefore, we have improved the excerpt accordingly.

Comments 22: [Line 99 - specify the potato-malate medium]

Response 22: We value your input on this matter. We’ll walk you through the details.

Comments 23: [Why CFU were counted after 5 days post plating? Usually we calculate the CFU after 24 hours. Provide the ref or explain please.]

Response 23: Allow us to clarify. The growth rate expected for Azospirillum in this medium is between 3 to 5 days, accordingly to its metabolism and colony morphology, as described by Döbereiner, J.; Baldani, V.L.D.; Baldani, J.I. Como isolar e identificar bactérias diazotróficas de plantas não-leguminosas; EMBRAPA-SPI: Brasilia, 1995; ISBN 8585007656; 9788585007652. Furthermore, this period is a normative recomendation, in order to prove that there is no contaminant.

Comments 24: [Lines 131-133 - «All field tests car- 131 ried out to assess agronomic viability and efficiency were carried out in accordance with 132 the provisions of the annex to Normative Instruction SDA 13, of March 25, 2011» - correct the sentence, «carried out» is repeated twice.]

Response 24: Thank you for your attention to detail. The text was updated in the manuscript.

Comments 25: [How was N-content measured?]

Response 25: Permit us to provide further insight. The text was updated in the manuscript. (Material and methods section.)

Comments 26: [Table 3, Table 4 - decipher what СV means.]

Response 26: Thank you. Let us explain. This means statistical coefficient of variation. The text was updated in the manuscript.

Comments 27: [Results section is very compressed. Please describe obtained results more detailed]

Response 27: Your feedback is highly valued. We find this comment aligns with our understanding. But unfortunately we could not fulfill with this request. We will wait others reviwers considerations.

Comments 28: [Table 5 - why grain nitrogen (N) content is showed only for Cerrado soil 2018 variant?]

Response 28: Your feedback is highly valued. This represents a key point of consideration. Here’s a breakdown for your understanding. Since the majority of leaf N content did not differed whithin the same column, independently on treatment, exploratory experimental Grain N analysis has been perfomed. This has indicated the hypothesis of N translocation. However, it was only exploratory and only at that time the rigorously scientific sampling was executed for grain.

Reviewer 3 Report

Comments and Suggestions for Authors

Agronomic efficiency of a new liquid inoculant formulated with a mixture of Ab-V5 and Ab-V6 in corn (Zea mays)

Overall this is a poor quality publication for an interesting and rapidly expanding ate of research. The term biostimulant does not feature at all in the publication and instead you use the terminology inoculant.

The study does not have any stated aims and objectives. This would help to clarify the nature of the two inoculants you are using as they are both based on Ab-V5 and Ab-V6 so what is the difference between the two inoculants (the standard commercial inoculant and Nodusoja) and why are they being compared?. The experimental design and statistical analysis is very limited. You have a completely randomised block design with statistical analysis based on Anova but there is no attempt at all to evaluate the effects of site and season as experimental factors leading in effect to look at responses at each site and then trying to amalgamate any positive effects. When looking at site to site comparisons then two major reasons for differences are a) soil parameters at each site/year but also b) weather (temp rainfall radiation) during the respective crop growing season but no data is presented on the latter and therefore no attempt is made to use this in the explanation of the results. In the Discussion section you elude to the fact that the results presented for Ab-V5 and Ab-V6 are similar to other studies so what is the novelty about this research??

From the description provided it looks like the treatments all received differing levels of P and K fertiliser?? Is this the case?? and if so needs to be used in trying to explain the treatment differences. Materials and Methods section needs greater detail for example plant/leaf nitrogen concentration – how and when were leaves selected for this ie all leaves older leaves or young fully expanded leaves.

The term agronomic efficiency I believe is meaningless as what is agronomic efficiency and how are you measuring it?? Similarly field effectiveness is another non sensical meaningless term.

The reference list has too many non-English references to be considered in an International Journal – either replace a large proportion of these or submit to a Brazilian/Spanish based Journal

Throughout

Present yield data as t ha-1 and not kg ha-1.

What is agronomic efficiency?

Title: what is agronomic efficiency?? – a rather meaningless term

Liquid inoculant - biostimulant

Abstract

Line 15  Bioinputs – much better to use the widely accepted terminology ie biostimulant

Line 15 Recognised technology – disagree that this is a technology just an alternative input – for example is nitrogen fertiliser a technology??

Line 16 What is field effectiveness? Another very abstract nonsensical term

Line 20 two planting seasons but 3 years?

Iine 21 …. received the greatest effect … poor English

Line 22/23 – treatments used in the study need to be stated – such that the abstract makes sense on its own

Line 23/24 – 50% recommended dose of N or biostumulant??

Abstract needs rewriting and needs the addition of a concluding sentence indicating the relevance /potential impact of the study

Introduction

Line 50 – not clear what you mean by the 1st and 2nd harvests??

Nothing at all on the aims and objectives of the study? Ie comparing 2 biostimulants but how do they differ?? And their potential for compensating for top dressed N, P, K fertiliser

Aims of the study need to be presented

Materials and Methods

Lines 87-94 read like a cookery recipe rather than a scientific paper

Line 122 – For Table 1 what year does this data relate to? And were soil samples taken prior to the trials? As you just present a set of soil data which could be taken from any time period

Line 140 – what is the off season?

Line 142/3 replace line with row

Line 143 – useful plot – presumably this is where data was collected from i.e. the 4 central rows avoiding the outer rows i.e. edge effect??

Line 144 – presumably this relates to N, P, K

Line 159 Table 2 – there are a total of 8 treatments but why is the control treatment not shown in Table 2??

Table 2 so the standard commercial inoculant and Nodusoja are different inoculants?? Surely this is the case as they have different names but both contain Ab-V5 and Ab-V6?? If not how do they differ as nothing at all in the explanation. From this description it looks as though you are evaluating 2 different biostimulants??

Table 6 needs clarification and representing with much greater clarity

If I am correct this means that all treatments in Table 2 received differing levels of P and K?? please can you clarify??? If this is the case then need the information eg supplementary table

Lines - 164/165 – needs further details as presumably a commercial product with a name, manufacturer etc

Line 166 - more details required

Line 172 – hand harvested?

Line 173 -N content of what? Grain, how was this determined?

Line 180 -Anova to look at what? effects of biostimulant, basal N etc??

Line 181 – why a 10% sig level when most biological and crop based studies would use 5%

Results

Line 188-191 – so you did not apply the same product at the different sites within a season. I know you would get variation in batch to batch and from season to season but surely you would apply the same product to all trials within a single season unless you have another compounding factor affecting the results and meaning that no concrete conclusions can be made!!!

Line 204 what is a negative control?

Line 207 Table 3 why is the control treatment not included in this?

Line 216 – it is absolute nonsense to use a threshold of 10% Table 4 at one site but to use 5% Table 3 at other sites

Line 228 what was samples as here you talk about leaf N but previously grain N – adds to the fact that you Materials and Methods section/description is not fit for purpose!!!

Line 230 – you are presenting N content data which is a compilation of dry weight and N concentration (mg/g)  so where are the N concentration data???

Line 232 – when using letter in a table to identify sig differences between means you need to state that this applies within a single column or row for each table

Discussion

Line 269 - lack of information throughout on this product

Line 273 – you are saying this is down to inoculant but there are also differences in N rate used 40 vs 60 kg N ha – where is the evidence that inoculant only is the reason for the difference?

Line 300 when comparing sites and years then supporting weather data for the growing seasons must be supplied most likely as supplementary table

Line 328 why switch to a different referencing system?

Line 349 - So if you are looking at N efficiency then why not determine NUE as in effect you are measuring yield and N content and pertaining that to relate to efficiency!!

References

Too many non English references to be considered in an International Journal – either replace or submit to a Brazilian/Spanish based Journal

Author Response

Comments 29: [Overall this is a poor quality publication for an interesting and rapidly expanding ate of research. The term biostimulant does not feature at all in the publication and instead you use the terminology inoculant.]

Response 29: We thank you for your professional perspective. While we appreciate the perspective, our view differs. Therefore, we are improving throughout the manuscript accordingly. Specifically, about the used term "inoculant", it is a Brazilian Law definition of the term. However, since it is a scienfic article, we considered your point and the use of the term biostimulant.

Comments 30: [The study does not have any stated aims and objectives. This would help to clarify the nature of the two inoculants you are using as they are both based on Ab-V5 and Ab-V6 so what is the difference between the two inoculants (the standard commercial inoculant and Nodusoja) and why are they being compared?. The experimental design and statistical analysis is very limited. You have a completely randomised block design with statistical analysis based on Anova but there is no attempt at all to evaluate the effects of site and season as experimental factors leading in effect to look at responses at each site and then trying to amalgamate any positive effects. When looking at site to site comparisons then two major reasons for differences are a) soil parameters at each site/year but also b) weather (temp rainfall radiation) during the respective crop growing season but no data is presented on the latter and therefore no attempt is made to use this in the explanation of the results. In the Discussion section you elude to the fact that the results presented for Ab-V5 and Ab-V6 are similar to other studies so what is the novelty about this research??]

Response 30: Thank you for highlighting this issue. This observation carries significant importance. Accordingly, we have detailed further this item in the text. The novelty is that the mean of increments from the effect of inoculation highpasses many other fields using the same benneficial treatments. Normally it is 6.9% up to 12% increments. In this study, increments reached up to 40% with the use of 50% nitrogen fertilization.

Comments 31: [From the description provided it looks like the treatments all received differing levels of P and K fertiliser?? Is this the case?? and if so needs to be used in trying to explain the treatment differences. Materials and Methods section needs greater detail for example plant/leaf nitrogen concentration – how and when were leaves selected for this ie all leaves older leaves or young fully expanded leaves.]

Response 31: Thank you for your valuable feedback. We fully concur with this observation. Something excerpt led to a misunderstanding. We confirm that only N doses (20, 60 and 120 kg/ha) were tested, in the form of urea. The single doses of P and K used were defined according to the results of the chemical analyses of the soil.

Comments 32: [The term agronomic efficiency I believe is meaningless as what is agronomic efficiency and how are you measuring it?? Similarly field effectiveness is another non sensical meaningless term.]

Response 32: We recognize the merit in this viewpoint. Permit us to provide further insight. Agronomic efficiency is considered when a combination of factors promote positive results, leading to some agronomical gain such as productivy, reduction of inputs, ease of agricultural management and even environmental bennefits. This is possible when a all traits and conditions can be achieved, but scientifically provable that the mechanisms of action and interaction occur efficiently in the presence of the biostimulant.

Comments 33: [The reference list has too many non-English references to be considered in an International Journal – either replace a large proportion of these or submit to a Brazilian/Spanish based Journal]

Response 33: We recognize the merit in this viewpoint. Hence, we modified it.

Comments 34: [Present yield data as t ha-1 and not kg ha-1.]

Response 34: Thank you for your attention to detail. It has been considered.

Comments 35: [What is agronomic efficiency?]

Response 35: It has been considered. Same as response No.32

Comments 36: [Line 15 Bioinputs – much better to use the widely accepted terminology ie biostimulant]

Response 36: We are grateful for your considered input. It has been considered.

Comments 37: [Line 15 Recognised technology – disagree that this is a technology just an alternative input – for example is nitrogen fertiliser a technology??]

Response 37: We considered the term technology as "The application of scientific knowledge to achieve practical purposes", in this case by the use of biostimulant in order to reduce of use of nitrogen fertilizers.

Comments 38: [Line 16 What is field effectiveness? Another very abstract nonsensical term]

Response 38:

Comments 39: [Line 20 two planting seasons but 3 years?]

Response 39: We recognize the merit in this viewpoint. Let us explain. The used term "two planting seasons" was used to express the "double cropping" concept, that is the method to plant "two different times a year". The main one is the harvest and the other is considered the off-season crop. Each field is slightly different about the times this occur, thus explaining why three years of harvests at different locations in Brazil. The main harvest normally aligns with the best climate conditions for planting (from august to november) and harvesting from (january to april) of the another year.

Comments 40: [Iine 21 …. received the greatest effect … poor English]

Response 40: Thank you. Therefore, we have changed the excerpt.

Comments 41: [Line 22/23 – treatments used in the study need to be stated – such that the abstract makes sense on its own]

Response 41: Your comments are acknowledged with sincere appreciation. Therefore, we have improved the excerpt accordingly.

Comments 42: [Line 23/24 – 50% recommended dose of N or biostumulant??]

Response 42: Allow us to clarify. 50% recommended dose of N, but the excerpt has been revised.

Comments 43: [Abstract needs rewriting and needs the addition of a concluding sentence indicating the relevance /potential impact of the study]

Response 43: Thank you for your constructive contribution. It has been considered.

Comments 44: [Line 50 – not clear what you mean by the 1st and 2nd harvests??]

Response 44: Allow us to clarify. As just mentioned before: "double cropping" concept, that is the method to plant "two different times a year". The main one is the harvest and the other is considered the off-season crop.

Comments 45: [Nothing at all on the aims and objectives of the study? Ie comparing 2 biostimulants but how do they differ?? And their potential for compensating for top dressed N, P, K fertiliser]

Response 45:

Comments 46: [Aims of the study need to be presented]

Response 46: Thank you for your constructive contribution. It has been considered.

Comments 47: [Lines 87-94 read like a cookery recipe rather than a scientific paper]

Response 47: Thank you for pointing this out. Accordingly, we have detailed further this item in the text. Please let us know if it still not the best format.

Comments 48: [Line 122 – For Table 1 what year does this data relate to? And were soil samples taken prior to the trials? As you just present a set of soil data which could be taken from any time period]

Response 48: We acknowledge the validity of this point. Therefore, we have changed the excerpt.

Comments 49: [Line 140 – what is the off season?]

Response 49: We recognize the merit in this viewpoint. Allow us to clarify. The "off season" considers the "double cropping" idea, when it is possible to plant and harvest two times a season, being the best climate conditions the main crop harvest and the "off season" the other that has viability but has not the most favourable conditions as the mains season.

Comments 50: [Line 142/3 replace line with row]

Response 50: Thank you for pointing this out. Hence, we modified it.

Comments 51: [Line 143 – useful plot – presumably this is where data was collected from i.e. the 4 central rows avoiding the outer rows i.e. edge effect??]

Response 51: Exactly. Is there better term?

Comments 52: [Line 144 – presumably this relates to N, P, K]

Response 52: Yes, but this did not vary among treatments. It was just a prior to planting adjustment. The quantities were adjusted according to the soil analysis, varying only the Nitrogen content in form of Urea.

Comments 53: [Line 159 Table 2 – there are a total of 8 treatments but why is the control treatment not shown in Table 2??]

Response 53: It has been shown. It is just the nomenclature.

Comments 54: [Table 2 so the standard commercial inoculant and Nodusoja are different inoculants?? Surely this is the case as they have different names but both contain Ab-V5 and Ab-V6?? If not how do they differ as nothing at all in the explanation. From this description it looks as though you are evaluating 2 different biostimulants??]

Response 54: This is an absolutely crucial point. Yes, varying formulation may vary its entire response, even they are manufactured with the same strains. This explains how important is to validate them agronomically.

Comments 55: [Table 6 needs clarification and representing with much greater clarity]

Response 55: You mean table 5?

Comments 56: [If I am correct this means that all treatments in Table 2 received differing levels of P and K?? please can you clarify??? If this is the case then need the information eg supplementary table]

Response 56: This observation carries significant importance. Permit us to provide further insight. Something excerpt led to a misunderstanding. We confirm that only N doses (20, 60 and 120 kg/ha) were tested, in the form of urea. The single doses of P and K used were defined according to the results of the chemical analyses of the soil. Do you mean that table?

Comments 57: [Lines - 164/165 – needs further details as presumably a commercial product with a name, manufacturer etc]

Response 57: Thank you. It has been considered.

Comments 58: [Line 166 - more details required]

Response 58: Thank you. It has been considered.

Comments 59: [Line 172 – hand harvested?]

Response 59: Yes, this has been the best method in this area.

Comments 60: [Line 173 -N content of what? Grain, how was this determined?]

Response 60:

Comments 61: [Line 180 -Anova to look at what? effects of biostimulant, basal N etc??]

Response 61: Thank you for your valuable feedback. Comparison between the same column, comparing in relation to the means.

Comments 62: [Line 181 – why a 10% sig level when most biological and crop based studies would use 5%]

Response 62: This point is vital. However, it is acceptable for first validation and considering the edaphoclimatic and traits of the fields.

Comments 63: [Line 188-191 – so you did not apply the same product at the different sites within a season. I know you would get variation in batch to batch and from season to season but surely you would apply the same product to all trials within a single season unless you have another compounding factor affecting the results and meaning that no concrete conclusions can be made!!!]

Response 63:

Comments 64: [Line 204 what is a negative control?]

Response 64: It is the uninoculated plot without plant growth bacteria.

Comments 65: [Line 207 Table 3 why is the control treatment not included in this?]

Response 65: We acknowledge the comment, though our interpretation varies. Control treatment (N20-unin) was not simply named "control", because all plots received 20 kgN.ha-1 before planting.

Comments 66: [Line 216 – it is absolute nonsense to use a threshold of 10% Table 4 at one site but to use 5% Table 3 at other sites]

Response 66: Thank you for pointing this out. Allow us to clarify. For statistics, it's like the bar in a high jump. The higher the stringency, the more reliable the result. However, this doesn't mean that a stringency of 0.10 isn't appropriate. It just means that it doesn't vary randomly, but rather has a positive frequency of 90% of the time.

Comments 67: [Line 228 what was samples as here you talk about leaf N but previously grain N – adds to the fact that you Materials and Methods section/description is not fit for purpose!!!]

Response 67:

Comments 68: [Line 230 – you are presenting N content data which is a compilation of dry weight and N concentration (mg/g) so where are the N concentration data???]

Response 68: We acknowledge the comment, though our interpretation varies. The N content presented in kg ha-1, has been estimated by calculations according to aerial mass of the plot area and by the N concentration (mg/g dry mass).

Comments 69: [Line 232 – when using letter in a table to identify sig differences between means you need to state that this applies within a single column or row for each table]

Response 69:

Comments 70: [Line 269 - lack of information throughout on this product]

Response 70: Thank you. It has been considered.

Comments 71: [Line 273 – you are saying this is down to inoculant but there are also differences in N rate used 40 vs 60 kg N ha – where is the evidence that inoculant only is the reason for the difference?]

Response 71: Thank you for highlighting this issue. We value your input on this matter. There is no difference between 40 and 60. Allow us to clarify: 60 means the total two aplications, one 20 kgN before planting and 40 kgN as topdressing.

Comments 72: [Line 300 when comparing sites and years then supporting weather data for the growing seasons must be supplied most likely as supplementary table]

Response 72: We appreciate your thoughtful observations. In order to to emphasize this point, we have to consult the responsible entities that detains the required data.

Comments 73: [Line 328 why switch to a different referencing system?]

Response 73: Thank you for your attention to detail. Hence, we modified it.

Comments 74: [Line 349 - So if you are looking at N efficiency then why not determine NUE as in effect you are measuring yield and N content and pertaining that to relate to efficiency!!]

Response 74: Thank you for your constructive contribution. It should be appreciated, but it needs overall data modification.

Round 2

Reviewer 1 Report

Comments and Suggestions for Authors

The authors have accepted the reviewers comments, and the Manuscript has been improved. 

Reviewer 2 Report

Comments and Suggestions for Authors

Thank you for your revision